# Neural Network-Based Adaptive Sigmoid Circular Path-Following Control for Underactuated Unmanned Surface Vessels under Ocean Disturbances

**Yi Ren [1,2], Lei Zhang [1,*], Wenbin Huang [1] and Xi Chen [1]**

1  Science and Technology on Underwater Vehicle Laboratory, Harbin Engineering University, Harbin 150001, China; renyi_708@163.com (Y.R.); huangwenbin@mail.nwpu.edu.cn (W.H.); chenxibruce@163.com (X.C.)
2  No.708 Research Institute of CSIC, Shanghai 200011, China
*  Correspondence: zhanglei103@hrbeu.edu.cn

**Abstract:** This study describes a circular curve path-following controller for an underactuated unmanned surface vessel (USV) experiencing unmodeled dynamics and external disturbances. Initially, a three degrees of freedom kinematic model of the USV is proposed for marine environmental disturbances and internal model parameter deterrence. Then, the circular path guidance law and controller are designed to ensure that the USV can move along the desired path. During the design process, a proportional derivative (PD)-based sigmoid fuzzy function is applied to adjust the guidance law. To accommodate unknown system dynamics and perturbations, a radial basis function neural network and adaptive updating laws are adopted to design the surge motion and yaw motion controllers, estimating the unmodeled hydrodynamic coefficients and external disturbances. Theoretical analysis shows that tracking errors are uniformly ultimately bounded (UUB), and the closed-loop system is asymptotically stable. Finally, the simulation results show that the proposed controller can achieve good control effects while ensuring tracking accuracy and demonstrating satisfactory disturbance rejection capability.

**Keywords:** path following; line of sight; robust control; underactuated unmanned surface vehicle; sliding mode; adaptive control; radial basis function neural networks

## 1. Introduction

Over the past few decades, unmanned surface vessels (USVs) have gained widespread attention due to their broad application potential in various marine missions, such as environmental monitoring, ocean exploration, and military tasks [1]. Among various USV control techniques, path-following control is a fundamental technology that enables the basic autonomous sailing ability for USVs in ocean missions, e.g., maritime surveillance, seabed charting, and environment monitoring. To realize the path-following objective for USVs, various research efforts have been made by researchers, i.e., [2–7].

The control objective of a path-following control system is to steer an USV along a time-independent trajectory under internal model dynamics and external disturbances [8]. Generally speaking, the path-following controller of underactuated USVs is always constructed using two components: the path-following guidance law and the dynamic tracking law. First, the path-following guidance law generates a desired yaw angle to guide the underactuated USV to converge to the desired path. Then, the dynamic tracking law forces the USV to converge to the desired speed and the guided yaw angle. In existing studies, widely adopted approaches include the line-of-sight (LOS) (e.g., [9–11]) and vector field (VF) methods [12–14]. Compared with VF-based methods, the LOS technique has garnered widespread research focus given its simpler and more reliable algorithm as well as better transient performance. The conventional LOS guidance approach was established by [10].

Here, a look-ahead distance is introduced to determine the LOS point, and the desired LOS angle is generated by the relative direction between the USV position and the LOS coordinates. Although effective, it should be pointed out that the method developed by [10] is only based on a fixed look-ahead distance. As a consequence, the system robustness under external disturbances and guidance performance under different speeds cannot be guaranteed. For the first issue, the integral LOS approach represents an effective solution, as first proposed by [11]. The main idea of the integral LOS is to introduce the integration of cross-tracking error into the guidance law to enhance the steady-state guidance performance under constant ocean disturbances. Under this condition, ref. [15] proposes a relative velocity model to simplify the control system under constant ocean current, which realizes the global asymptotic stability and local exponential stability of the guidance system. In [16], a predictor-based LOS guidance technique was developed to compensate for the sideslip angle of the USV that is induced by disturbances. In [17], a fuzzy observer is introduced to assist the guidance of the USV's surge speed and yaw angle under multiple lumped unknowns. To address the second issue, an improved LOS approach is proposed by [18], in which the look-ahead distance is designed by considering the USV's speed and the cross-tracking error, such that both the system performance and speed adaptability of parameters can be guaranteed. In [19], the fuzzy optimizer is introduced to tune the look-ahead distance parameter. Ref. [20] proposed an adaptive line-of-sight (ALOS) algorithm that adapts to error variations and is applied for path following. Ref. [21] proposed a full-speed adaptive guiding law to solve the standard LOS systems' lack of sensitivity to lateral errors. However, it should be pointed out that most of the above-mentioned methods are only effective in straight-line path guidance. For curved path tracking, the Serret-Frenet coordinate system is required to modify curve paths for tracking [22] or to estimate curved paths using monotone Hermite spline curves for parameterizing [23] linear paths. To enable LOS steering on a curve path, ref. [24] linearized the curved trajectory. Ref. [25] used the closest point on the trajectory to steer the USV as the LOS reference. Based on the Serret-Frenet coordinate transformation, ref. [26] further extended the fuzzy LOS guidance approach proposed by [17] to the curved path case. Although the above guidance algorithm can solve the guidance robustness under external disturbances with adaptability, it also refers to a complicated structure and poses a large computational burden in practice. For example, the above interference compensation method needs to identify a large number of fuzzy rules or update the weights of network nodes, which cannot meet the real-time requirements in engineering applications. For curve tracking, the related Serret-Frenet coordinate transformation may further expand the architecture of the guidance algorithm.

In addition the path guidance issue, the dynamic tracking issue is another key aspect that should be considered in realizing path-following control. In practice, a USV must maneuver through maritime disturbances, such as wind, currents, and unpredictable waves. Underactuated USVs may experience substantial velocity drift angles due to the influence of ocean currents, which makes it challenging for the actual trajectory to converge in the intended direction. In addition, due to the complicated geometric shape of the USV's hull and variable surrounding flow field, it is difficult to account for the hydrodynamic forces in the controller design. If unmodeled dynamics cannot be compensated for in these systems, the control performance will deteriorate and even cause instability. The above LOS techniques can significantly affect guiding stability in practical applications, which can result in substantial oscillations in the intended trajectory while only slightly improving performance. Furthermore, many control methods have been investigated to make the USV follow the desired yaw angle and surge velocity. In recent years, adaptive backstepping technology has drawn a lot of interest among these control algorithms because of its pronounced advantages in managing ocean disturbances and model uncertainty. To solve the problem of the path-following control of a USV, ref. [27] combined the backstepping method with LOS guidance to allow underactuated USVs to track arbitrary straight and curved trajectories. Fuzzy rules [28], neural networks [29], and other adaptive techniques are typically incorporated into the backstepping controller design process when taking

into account ocean disturbances and unmodeled dynamics to estimate and account for the model's unknown terms and increase the controller's robustness. It should be emphasized that only theoretical and simulation studies have been used to verify the approaches indicated above. There are still some restrictions in real-world engineering applications.

Motivated by the above observations, this study proposes a novel sigmoid function-based line-of-sight circular path guidance law and neural network-based path-following controller. The innovations of this study are as follows:

1.  A novel look-ahead angle guidance architecture is established to facilitate the circular path guidance. Compared with the existing curve path guidance approach, the Serret-Frenet coordinate transformation is not required by the proposed method, such that the proposed method enjoys a simple and intuitive structure as a traditional straight-line guidance approach in practical engineering.
2.  A novel sigmoid compensation is introduced to the guidance law. Using this design, the guidance angle can be adaptively tunned according to the scale and the change rate of cross-tracking error, which enables guidance performance under external disturbances and parameter adaptability under different surge speeds.
3.  A neural network-based adaptive scheme is developed to estimate unknown un-modeled dynamics. In the design, an adaptive learning law is developed to update the weight matrix of the neural network, and the adaptive control is combined to compensate for the deficiencies of the sliding mode change structure control so that the system can weaken the vibrations while maintaining the robustness under internal perturbations and external disturbances.

The remainder of this paper is organized as follows. The USV model and preliminary findings are introduced in Section 2. The proportion-based line-of-sight circular path guidance law is given in Section 3. The neural network controller design is presented in Section 4. Numerical verification results are given in Section 5. Conclusions are drawn in Section 6.

## 2. Preliminary Findings

### 2.1. Underactuated USV Model

In order to express the motion of the USV, the Earth-fixed frame $\{X_E O_E Y_E\}$ and the body-fixed frame $\{X_B O_B Y_B\}$ are introduced, which are illustrated in Figure 1.

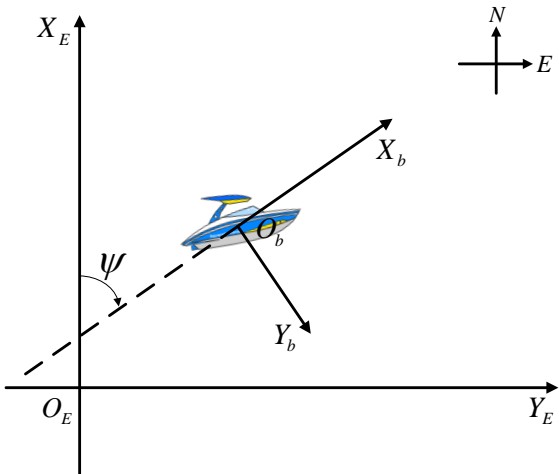

**Figure 1.** Coordinates of the USV motion.

In this article, we consider that the motions of the USV are defined in the horizontal plane. The USV model is given in Equation (1):

$$
\begin{cases}
\dot{x} = u\cos\psi - v\sin\psi \\
\dot{y} = u\sin\psi + v\cos\psi \\
\dot{\varphi} = r \\
\dot{u} = \frac{m_{22}}{m_{11}}vr - \frac{X_u}{m_{11}}u - \frac{X_{|v|v}}{m_{11}}|v|v - \frac{X_{|v|r}}{m_{11}}|v|r \\
\quad - \frac{X_{|r|r}}{m_{11}}|r|r + \frac{d_u}{m_{11}} + \frac{\tau_u}{m_u} \\
\dot{v} = -\frac{m_{11}}{m_{22}}ur - \frac{Y_v}{m_{22}}v - \frac{Y_{|v|v}}{m_{22}}|v|v - \frac{Y_{|v|r}}{m_{22}}|v|r \\
\quad - \frac{Y_{vvr}}{m_{22}}v^2r - \frac{Y_{vrr}}{m_{22}}vr^2 + \frac{d_v}{m_{22}} \\
\dot{r} = \frac{m_{11}-m_{22}}{m_{33}}uv - \frac{N_r}{m_{33}}r - \frac{N_{|v|r}}{m_{33}}|v|r - \frac{N_{|r|r}}{m_{33}}|r|r \\
\quad - \frac{N_{vvr}}{m_{33}}v^2r - \frac{N_{vrr}}{m_{33}}vr^2 + \frac{d_r}{m_{33}} + \frac{\tau_r}{m_r}
\end{cases}
\tag{1}
$$

where $x$, $y$ represent the USV's position; $\psi$ stands for the yaw angle; $u$, $v$, and $r$ stand for the surge velocity, sway velocity, and yaw velocity related to the body-fixed coordinate, respectively; $m_{11}$, $m_{22}$, and $m_{33}$ stand for inertia masses; $\tau_u$ and $\tau_r$ are control inputs provided by the equipped thrusters and rudders, respectively; $d_u$, $d_v$, and $d_r$ are the coupling of ocean disturbances; and $X_{(\bullet)}$, $Y_{(\bullet)}$, and $N_{(\bullet)}$ are constant hydrodynamic coefficients.

To facilitate descriptions in the subsequent controller design, we reorganize the above USV model to provide the following simplified form:

$$
\begin{cases}
\dot{x} = u\cos\psi - v\sin\psi \\
\dot{y} = u\sin\psi + v\cos\psi \\
\dot{\varphi} = r \\
\dot{u} = F_u + \frac{\tau_u}{m_u} \\
\dot{v} = F_v \\
\dot{r} = F_r + \frac{\tau_r}{m_r}
\end{cases}
\tag{2}
$$

where $m_u = m_{11}$ and $m_r = m_{33}$. The couplings of unknown nonlinear hydrodynamics and external disturbances $F_u$, $F_v$, and $F_r$ are expressed as follows:

$$
\begin{cases}
F_u = \frac{m_{22}}{m_{11}}vr - \frac{X_u}{m_{11}}u - \frac{X_{|v|v}}{m_{11}}|v|v - \frac{X_{|v|r}}{m_{11}}|v|r \\
\quad - \frac{X_{|r|r}}{m_{11}}|r|r + \frac{d_u}{m_{11}} \\
F_v = -\frac{m_{11}}{m_{22}}ur - \frac{Y_v}{m_{22}}v - \frac{Y_{|v|v}}{m_{22}}|v|v - \frac{Y_{|v|r}}{m_{22}}|v|r \\
\quad - \frac{Y_{vvr}}{m_{22}}v^2r - \frac{Y_{vrr}}{m_{22}}vr^2 + \frac{d_v}{m_{22}} \\
F_r = \frac{m_{11}-m_{22}}{m_{33}}uv - \frac{N_r}{m_{33}}r - \frac{N_{|v|r}}{m_{33}}|v|r - \frac{N_{|r|r}}{m_{33}}|r|r \\
\quad - \frac{N_{vvr}}{m_{33}}v^2r - \frac{N_{vrr}}{m_{33}}vr^2 + \frac{d_r}{m_{33}}
\end{cases}
\tag{3}
$$

**Remark 1.** *As a typical underactuated system, most of USV's actuators can only provide surge and yaw control forces, while dynamics in sway motion are underactuated. Thus, we only consider $\tau_u$ and $\tau_r$ to enhance the practicability of the proposed results. In addition, the hydrodynamics of USVs tend to exhibit a high degree of nonlinearity and perturbation, making it difficult to obtain accurate hydrodynamic coefficients. Therefore, couplings $F_u$, $F_v$, and $F_r$ that related to the USV's hydrodynamic characteristics will all treated as unmodeled dynamics when designing the controller, which is significant to improve the practicability of the controller. In the following controller design, a neural network approximator is designed to estimate these unknown terms.*

**Assumption 1.** *Unknown system dynamics included in $F_u$, $F_v$, and $F_r$ are upper bounded with unknown bounds.*

**Assumption 2.** *The USV's velocities $u$, $v$, and $r$ and accelerations $\dot{u}$, $\dot{v}$, and $\dot{r}$ have known bounds.*

**Remark 2.** *It is worth noting that Assumption 1 and Assumption 2 are generally assumed in the controller design of marine vehicles, as noted in [30–33]. In the marine environment, the energy of the external interference is limited, so Assumption 1 holds. In actual applications, the USV's actuators will also be limited by the finite energy (i.e., input saturation constraint). Therefore, Assumption 2 also holds in engineering. In addition, the maximum velocities and acceleration of the USV can be obtained by sea tests (i.e., speed test, turn test, and zigzag test).*

### 2.2. Function Approximation

In this study, the radial basis function neural network (RBFNN) is employed to approximate the unknown hydrodynamics coupling and external disturbances. The approximation ability of RBFNNs is given as follows:

**Lemma 1 [33].** *For the real continuous function $f(X_n)$, $\mathbb{R}^d \to \mathbb{R}$, it can be approximated by following NN over to any arbitrary accuracy:*

$$f(X_n) = W^T h(X_n) + \varepsilon \tag{4}$$

*where $W = [w_1, \ldots, w_m]^T$ is the RBFNN weight; $\varepsilon$ is the RBFNN approximation error; $X_n$ is the network input vector; and $h(X_n) = [h_1(X_n), \ldots, h_m(X_n)]$ describes the networks' hidden layer, which is specified by the following Gaussian activation function:*

$$h_j(X_n) = \exp\left(-\frac{\|x(X_n) - c_j\|^2}{2b_j^2}\right), \; j = 1, 2, \ldots, m \tag{5}$$

*where $c_j$, $j = 1, \ldots, m$ is the column vector representing the center distribution of $X_n$, and $b_j$ represents the width of $h_j(X_n)$.*

### 2.3. Control Objective

This study seeks to solve the circular path-following control problem for an underactuated USV under LOS guidance and a neural network-based adaptive control architecture. The control objective of this study can be summarized in the following two points:

(1)  LOS guidance: Under the proposed LOS guidance law, the USV can track the circular path, and the transversal deviation $S_E$ is stable within the bounded tracking error.

(2)  Yaw angle and velocity tracking: Under the proposed controller, the yaw tracking error and the velocity tracking error dynamics are stable, and the related tracking errors satisfies the bounded solution (See in [34,35]) under the unknown hydrodynamics and external disturbances.

To realize the above control objective, an LOS guidance-based and neural network controller is proposed, and its structure is presented in Figure 2.

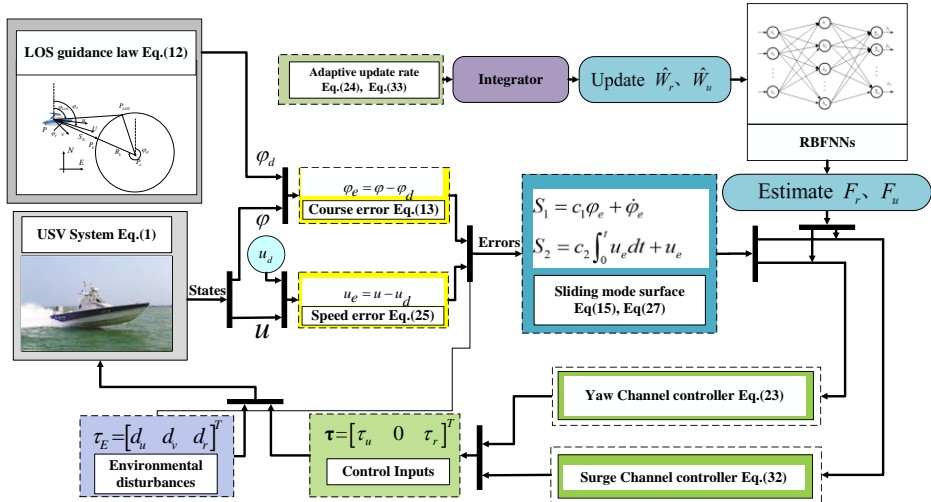

**Figure 2.** Schematic of the proposed LOS guidance and neural network controller for the USV.

## 3. Sigmoid Function-Based Line-of-Sight Circular Path Guidance Strategy

In this section, the sigmoid function-based LOS circular path guidance law is designed to facilitate the first control objective in the underactuated USV's path-following mission. For a better illustration, a schematic drawing of the designed guidance law is provided in Figure 3. In Figure 3, $P(x, y)$ is the position of the USV, $P_o(x_o, y_o)$ is the center of the tracked circle path, and $P_k(x_k, y_k)$ is the nearest intersection point of $P_oP$ and the tracked circle path. According to the above definitions, the rotation angle of $\overrightarrow{P_oP}$ can be calculated as follows:

$$\varphi_p = \text{atan2}\left(\frac{y - y_o}{x - x_o}\right) \tag{6}$$

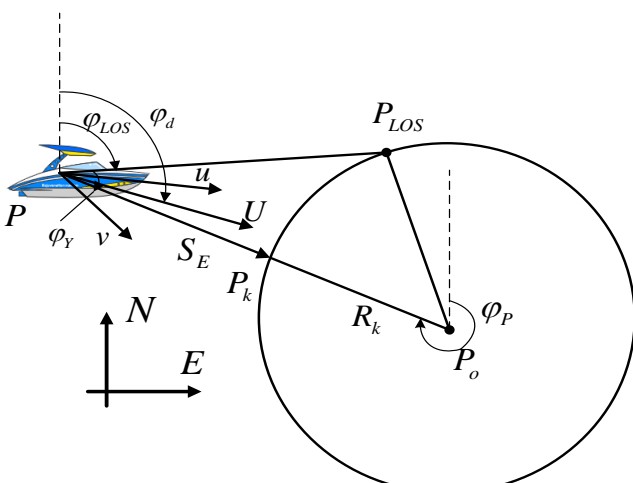

**Figure 3.** Line-of-sight-based circular path guidance model.

Inspired by the look-ahead distance in the straight-line guidance strategy, we introduce $d_t$ as the look-ahead angle in the circular path guidance strategy. Then, the LOS guidance point $P_{LOS}(x_{LOS}, y_{LOS})$ is described as follows:

$$\begin{cases} x_{LOS} = x_o + R_k \cos(\varphi_p + d_t) \\ y_{LOS} = y_o + R_k \sin(\varphi_p + d_t) \end{cases} \tag{7}$$

According to Equations (6) and (7), the LOS angle of the USV with respect to $P_{LOS}$ is given as follows:

$$\varphi_{LOS} = \text{atan2}\left(\frac{y_{LOS} - y}{x_{LOS} - x}\right) \tag{8}$$

To proceed with the sigmoid-based LOS circular path guidance law design, we denote the following auxiliary angle:

$$\varphi_Y = \left| \text{atan2}\left(\frac{y_o - y}{x_o - x}\right) - \text{atan2}\left(\frac{y_{LOS} - y}{x_{LOS} - x}\right) \right| \tag{9}$$

Then, in Equation (9), the expected yaw angle $\varphi_d$ of the USV is given as follows:

$$\varphi_d = \text{atan2}\left(\frac{y_{LOS} - y}{x_{LOS} - x}\right) + k_{S_E}\varphi_Y \tag{10}$$

where $k_{S_E}$ is the adaptive parameter, which is given by the following sigmoid function:

$$k_{S_E} = \frac{2}{1 + \exp(-k_1 S_E - k_2 \dot{S}_E)} - 1 \tag{11}$$

where $k_1$ and $k_2$ are positive design parameters, and $S_E$ is the path-following error, which is given as follows:

$$S_E = \sqrt{(y_o - y)^2 + (x_o - x)^2} - R_k \tag{12}$$

**Remark 3.** *The proportion-based line-of-sight circular path guidance can track the circular path. In addition, it has full-speed adaptability and strong robustness, and the parameters are easily set. It mainly includes the following aspects:*

(1) *The conventional line-of-sight method is suitable only for straight-line tracking and not for curve tracking. The existing curve LOS method needs to convert the system coordinate into the Serret-Frenet coordinate system when conducting curve path tracking. However, the LOS method proposed here does not require conversion, so the algorithm is more intuitive and practical.*

(2) *The compensation item $k_{S_E}$ is introduced to ensure the convergence performance under different $S_E$ values. When $S_E$ is large, the compensation item is large, which ensures the transient guidance performance and allows the USV to converge to the path faster. When $S_E$ is small, the response compensation becomes smaller to avoid oscillation.*

(3) *The curved path LOS guidance law is based on the traditional LOS guidance principle but includes the overall measures to deal with environmental interference. The new guidance law overcomes the disadvantages of the traditional sight law, which is vulnerable to environmental interference, while retaining the intuition and simplicity of the traditional line-of-sight guidance.*

## 4. Neural Network-Based Control Law

Based on the desired heading angle generated by the guidance law designed in Section 3, this section aims to design a heading speed controller based on neural networks and sliding mode adaptive technology that enables the USV to follow the desired path. In the design process of the controller, we introduce the RBFNN to approximate the sum of the internal and external disturbances of the system. At the same time, combined with the good effect of the sliding mode control strategy in nonlinear control, a neural network sliding mode adaptive controller is designed. Here, the adaptive update rate is used to adjust the weight matrix of the RBFNN in real time, and the sliding mode surface is used to enhance the stability of the system. The design of the control strategy is divided into two aspects: the yaw motion controller and the surge motion controller.

*4.1. Control Law Design*

4.1.1. Yaw Motion Controller Design

In this subsection, the yaw tracking control law is designed to force $\varphi$ to converge to its desired value $\varphi_d$, which is generated by the above designed guidance law. We define the yaw angle error $\varphi_e$ as follows:

$$\varphi_e = \varphi - \varphi_d \tag{13}$$

By differentiating Equation (13), the derivative of yaw angle error can be obtained:

$$\dot{\varphi}_e = r - \dot{\varphi}_d \tag{14}$$

Then, we define the sliding mode surface as follows:

$$S_1 = c_1 \varphi_e + \dot{\varphi}_e \tag{15}$$

where $c_1$ is the positive control parameter.

By differentiating Equation (15), the derivative of the sliding mode surface can be obtained:

$$
\begin{aligned}
l\dot{S}_1 &= c_1 \dot{\varphi}_e + \ddot{\varphi}_e \\
&= c_1 \dot{\varphi}_e + \dot{r} - \ddot{\varphi}_d
\end{aligned} \tag{16}
$$

According to Equation (2), we can rewrite the USV's yaw motion dynamics as follows:

$$
\begin{cases}
\dot{\varphi} = r \\
\dot{r} = F_r + G_r \tau_r
\end{cases} \tag{17}
$$

where $G_r = 1/m_r$ is defined for the sake of compactness.

By substituting Equation (17) into Equation (16), we obtain the following:

$$\dot{S}_1 = c_1 \dot{\varphi}_e + F_r + G_r \tau_r - \ddot{\varphi}_d \tag{18}$$

Unknown model dynamics $F_r$ exist in Equation (18). To facilitate the subsequent design, the following RBFNN can be constructed to approximate $F_r$ under Lemma 1:

$$F_r = W_r^T h_r(X) + \varepsilon_r \tag{19}$$

where $W_r = [w_{r1}, \ldots, w_{rm}]^T$ is the RBFNN's weight vector; $h_r = [h_{r1}, \ldots, h_{rm}]^T$ is the output of the network hidden layer; and $|\varepsilon_r| \leq \varepsilon_N$ is the bounded approximation error when $\varepsilon_N$ is a small positive constant. The input vector of the above NN is given as $X = [u, v, r]$.

Since $W_r$ is the unknown desired weight vector, we define $\hat{W}_r$ as its estimation. Then, the NN estimation error can be defined as follows:

$$\widetilde{W}_r = W_r - \hat{W}_r \tag{20}$$

Using the constant property of $W_r$, the time derivative of $\widetilde{W}_r$ can be obtained as follows:

$$
\begin{aligned}
\dot{\widetilde{W}}_r &= \dot{W}_r - \dot{\hat{W}}_r \\
&= -\dot{\hat{W}}_r
\end{aligned} \tag{21}
$$

Then, based on the above designs, the control law of the yaw motion channel is proposed as follows:

$$\tau_r = \frac{1}{G_r}(-c_1 \dot{\varphi}_e - k_r S_1 - \hat{W}_r^T h_r + \ddot{\varphi}_d) \tag{22}$$

where $k_r$ is the positive control parameter. The updated law of $\hat{W}_r$ is proposed as follows:

$$\dot{\hat{W}}_r = \gamma_{r1}(S_1 h_r(x) - \gamma_{r2}\hat{W}_r) \tag{23}$$

where $\gamma_{r1}$ and $\gamma_{r2}$ are positive control parameters.

4.1.2. Surge Motion Controller Design

In this subsection, the surge motion controller is designed to allow the USV's surge speed $u$ to converge to the target speed $u_d$, which is given by the planner. The velocity tracking error of USV can be defined as follows:

$$u_e = u - u_d \tag{24}$$

By differentiating Equation (24), the derivative of velocity error can be obtained:

$$\dot{u}_e = \dot{u} - \dot{u}_d \tag{25}$$

Then, we construct the integral sliding mode surface as follows:

$$S_2 = u_e + c_2 \int_0^t u_e(\tau)dt \tag{26}$$

where $c_2$ is the positive control parameter.

By differentiating Equation (26) along with Equation (2), the derivative of the sliding mode surface can be obtained:

$$\begin{aligned} \dot{S}_2 &= \dot{u}_e + c_2 u_e \\ &= c_2 u_e + \dot{u} - \dot{u}_d \\ &= c_2 u_e + F_u + G_u \tau_u - \dot{u}_d \end{aligned} \tag{27}$$

where $G_u = 1/m_u$.

Similar to the design in the yaw tracking channel, according to Lemma 1, we can also construct the following RBFNN to realize the approximation of $F_u$, which is formed using the following:

$$F_u = W_u{}^T h_u(x) + \varepsilon_u \tag{28}$$

where $W_u = [w_{u1}, \ldots, w_{um}]^T$ is the RBFNN's weight vector; $h_u = [h_{u1}, \ldots, h_{um}]^T$ is the output of the network hidden layer; $\varepsilon_u$ is the RBFNN's approximation error satisfying $|\varepsilon_u| \le \varepsilon_X$, where $\varepsilon_X$ is a small bounded constant; and $X = [u, v, r]^T$ is the network input vector.

Here, $\hat{W}_u$ is defined as the estimated value of $W_u$, and the estimated error can be expressed as follows:

$$\widetilde{W}_u = W_u - \hat{W}_u \tag{29}$$

By differentiating Equation (29), the derivative of estimated error can be obtained:

$$\begin{aligned} \dot{\widetilde{W}}_u &= \dot{W}_u - \dot{\hat{W}}_u \\ &= -\dot{\hat{W}}_u \end{aligned} \tag{30}$$

where the property $\dot{W}_u = 0$ is utilized in obtaining Equation (31).

Then, based on the above designs, the control law of the surge motion channel is proposed as follows:

$$\tau_u = \frac{1}{G_u}(-c_2 u_e - \hat{W}_u^T h_u + \dot{u}_d - k_u S_2) \tag{31}$$

where $k_u$ is a positive control parameter. The updated law of $\hat{W}_u$ is proposed as follows:

$$\dot{\hat{W}}_u = \gamma_{u1}(S_2 h_u(x) - \gamma_{u2}\hat{W}_u) \tag{32}$$

where $\gamma_{u1}$ and $\gamma_{u2}$ are positive control parameters.

**Remark 4.** *The RBFNN sliding mode adaptive controller designed in this section can be used to solve the problems of unknown disturbance and model parameter uncertainties, and the combination of the neural network and sliding mode makes the control system have better robustness and stability. It mainly includes the following aspects:*

1   *Neural networks have the ability to learn arbitrary functions, and its self-learning ability can avoid complex mathematical analysis that occupies an important position in traditional adaptive control theory. Aiming at solving the highly nonlinear control problems that cannot be solved by traditional control methods, the hidden neurons of multi-layer neural networks adopt an activation function with a nonlinear mapping function, which can approximate arbitrary nonlinear functions, providing an effective solution for nonlinear control problems. This method can be widely used to solve control problems with uncertain models.*

2   *The sliding mode control is combined with the neural network to approximate the nonlinear control system, and the neural network is used to realize the adaptive approximation of the internal and external disturbances of the system, which can effectively reduce the fuzzy gain. The adaptive law of neural networks is derived from the Lyapunov function, and the stability and convergence of the entire closed-loop system are ensured through the adjustment of adaptive weights.*

*4.2. Stability Analysis*

**Theorem 1.** *Considering the USV system described by Equation (2) and under the proposed control laws noted in Equations (22) and (31) and adaptive laws described in Equations (23) and (32), if Assumption 1 holds, the yaw tracking error $\varphi_e$ and the velocity tracking error $u_e$ are ultimately uniformly bounded (UUB).*

**Proof of Theorem 1.** To verify the stability of the closed-loop system, we consider the following Lyapunov function:

$$V = \frac{1}{2}S_1{}^2 + \frac{1}{2}S_2{}^2 + \frac{1}{2\gamma_{r1}}\widetilde{W}_r{}^T\widetilde{W}_r + \frac{1}{2\gamma_{u1}}\widetilde{W}_u{}^T\widetilde{W}_u \tag{33}$$

Noticing that $\gamma_{r1}$ and $\gamma_{u1}$ are all positive constant control parameters, it can be straight forward to verify that $V \geq 0$ and monotonically increases with respect to $S_1$, $S_2$, $\|\widetilde{W}_r\|$ and $\|\widetilde{W}_u\|$. To proceed with the verification, we can differentiate Equation (33) along with Equations (18) and (27). In this sequence, $\dot{V}$ can be obtained as follows:

$$\begin{aligned}
\dot{V} &= S_1\dot{S}_1 + S_2\dot{S}_2 - \frac{1}{\gamma_{r1}}\widetilde{W}_r{}^T\dot{\hat{W}}_r - \frac{1}{\gamma_{u1}}\widetilde{W}_u{}^T\dot{\hat{W}}_u \\
&= S_1(c_1\dot{\varphi}_e + F_r + G_r\tau_r - \ddot{\varphi}_d) + S_2(c_2 u_e + F_u + G_u\tau_u - \dot{u}_d) \\
&\quad - \frac{1}{\gamma_{r1}}\widetilde{W}_r{}^T\dot{\hat{W}}_r - \frac{1}{\gamma_{u1}}\widetilde{W}_u{}^T\dot{\hat{W}}_u
\end{aligned} \tag{34}$$

By substituting control laws in Equations (22) and (31) and adaptive laws in Equations (23) and (32) into Equation (34), we obtain the following:

$$\begin{aligned}
\dot{V} &= -k_r S_1^2 + \varepsilon_r S_1 - k_u S_2{}^2 + \varepsilon_u S_2 \\
&\quad - \widetilde{W}_r{}^T(\frac{1}{\gamma_{r1}}\dot{\hat{W}}_r - S_1 h_r(x)) - \widetilde{W}_u{}^T(\frac{1}{\gamma}\dot{\hat{W}}_u - S_2 h_u(x)) \\
&= -k_r S_1^2 - k_u S_2{}^2 + \varepsilon_r S_1 + \varepsilon_u S_2 + \gamma_{r2}\widetilde{W}_r{}^T\hat{W}_r + \gamma_{u2}\widetilde{W}_u{}^T\hat{W}_u
\end{aligned} \tag{35}$$

Here, $\widetilde{W}_p = W_p - \hat{W}_p$ and $p = u, r$. Thus, using Young's inequality, the following relation holds:

$$
\begin{aligned}
\widetilde{W}_p{}^T \hat{W}_p &= \widetilde{W}_p{}^T \left( W_p - \widetilde{W}_p \right) \\
&\leq -\frac{1}{2} \widetilde{W}_p{}^T \widetilde{W}_p + \frac{1}{2} W_p{}^T W_p
\end{aligned}
\tag{36}
$$

In addition to using Young's inequality, we also obtain the following:

$$
\varepsilon_r S_1 \leq \frac{1}{2} \varepsilon_r^2 + \frac{1}{2} S_1^2
\tag{37}
$$

$$
\varepsilon_u S_2 \leq \frac{1}{2} \varepsilon_u^2 + \frac{1}{2} S_2^2
\tag{38}
$$

Then, by combining Equations (35)–(38), $\dot{V}$ can be developed as follows:

$$
\begin{aligned}
\dot{V} &\leq -k_r S_1^2 - k_u S_2^2 + \frac{1}{2} S_1^2 + \frac{1}{2} S_2^2 - \frac{1}{2} \gamma_{r2} \widetilde{W}_r{}^T \widetilde{W}_r - \frac{1}{2} \gamma_{u2} \widetilde{W}_u{}^T \widetilde{W}_u \\
&\quad + \frac{1}{2} \varepsilon_r^2 + \frac{1}{2} \varepsilon_u^2 + \frac{1}{2} \gamma_{r2} W_r{}^T W_r + \frac{1}{2} \gamma_{u2} W_u{}^T W_u \\
&= -KV + \Delta
\end{aligned}
\tag{39}
$$

where $K = \min\{2k_r - 1, 2k_u - 1, \gamma_{r1}\gamma_{r2}, \gamma_{u1}\gamma_{u2}\}$ and $\Delta = 0.5(\varepsilon_r^2 + \varepsilon_u^2 + \gamma_{r2} W_r{}^T W_r + \gamma_{u2} W_u{}^T W_u)$. Then, the integration of Equation (39) yields the following:

$$
0 \leq V(t) \leq \frac{\Delta}{K} + [V(0) - \frac{\Delta}{K}]e^{-Kt}
\tag{40}
$$

By observing Equation (40), one can deduce that $V(t)$ is UUB to the residual set $\Omega_V : [0, \frac{\Delta}{K}]$. Moreover, according to the definition of $V$ in Equation (33), we can further conclude that the closed-loop signals $S_1$, $S_2$, $\|\widetilde{W}_u\|$, and $\|\widetilde{W}_r\|$ are all UUB to the residual set $\Omega_1 : [0, \sqrt{2\Delta/K}]$, $\Omega_2 : [0, \sqrt{2\Delta/K}]$, $\Omega_3 : [0, \sqrt{2\gamma_{u1}\Delta/K}]$, and $\Omega_4 : [0, \sqrt{2\gamma_{r1}\Delta/K}]$, respectively. In addition, after $S_1$ and $S_2$ converge to a small neighborhood of the equilibrium point, the convergence of the yaw tracking error signals $\varphi_e$ and $u_e$ can be realized. This completes the proof. $\square$

## 5. Simulation Verification

In this section, numerical simulations are conducted to verify the effectiveness and robustness of the above-designed LOS guidance law and RBFNN-based adaptive controller.

### 5.1. System Configuration

In this section, a 7-meter planning boat is used as the simulation platform, and the mass coefficient and dimensionless hydrodynamic coefficient information of the model are given in Table 1. The factorization coefficient is $H = \frac{1}{2}\rho L d V^2$. Here, $\rho$ is the fluid density, $L$ is the ship length, $d$ is the average draft, and $V$ is the 2-norm of the USV's velocity vector.

**Table 1.** The USV's model parameters.

| Mass Coefficient | Hydrodynamic Coefficient | | |
|---|---|---|---|
| | **First Order** | **Second Order** | **Third Order** |
| $m_u = 4050.67$, $m_v = 4070.32$, $m_r = 14{,}451.12$ | $Y_v = -0.47155$, $Y_r = -0.0716$, $N_v = -0.1459$, $N_r = -0.0557$ | $X_{vv} = 0.0682$, $X_{vr} = -0.0025$, $X_{rr} = 0.0039$, $Y_{vv} = -0.4757$, $Y_{vr} = -0.2900$, $Y_{rr} = -0.035$, $N_{vv} = 0.0148$, $N_{vr} = 0.002$, $N_{rr} = -0.0401$ | $Y_{vvr} = -0.51445$, $Y_{vrr} = -1.72674$, $N_{vvr} = -0.34044$, $N_{vrr} = -0.02524$ |

In simulations, parameters of the guidance law are set as $d_t = 5\deg$, $k_1 = 0.05$, and $k_2 = 0.06$. Parameters of the yaw angle controller are set as $c_1 = 1$, $k_r = 0.05$, $\gamma_{r1} = 0.0001$, and $\gamma_{r2} = 0.00003$. Parameters of the velocity tracking controller are set as $c_2 = 1$,

$k_u = 0.013$, $\gamma_{r2} = 0.00003$, and $\gamma_{u1} = 0.000001$. Parameters of RBFNNs are set as $m = 30$ and $b_j = 0.6$, and $c_i$ is evenly spaced on $[-20, 20] \times [-5, 5] \times [-2, 2]$.

In order to verify the robustness of the proposed control scheme, the ocean waves, winds, and currents are considered in simulations, and the simulation methods of waves and winds are given in [36] and [37], respectively. Three sets of ocean disturbances are adopted for comparison, which are given in Table 2. All directions of the external disturbance are set as 45 degrees.

**Table 2.** Scenarios of ocean disturbances with their corresponding sea state level.

| Scenarios | Sea State Level | Average Wave Height (m) | Wind Speed (kn) | Current Speed (kn) |
|-----------|-----------------|--------------------------|-----------------|---------------------|
| I | Claim water | 0 | 0 | 0 |
| II | Level 1 | 0.05 | 1 | 3 |
| III | Level 3 | 0.88 | 8 | 4 |

*5.2. Simulation Results*

Simulation results are shown in Figures 4–7. Figure 4 shows trajectories and the transverse deviation $S_E$ of the USV under three scenarios. As shown in Figure 4, the path-following error of the USV is less than 10 m for all three scenarios. The mean square error (RMS) is calculated as an index to measure the performance of the path following. The RMS values in the three different scenarios were 2.37 m, 5.63 m, and 8.96 m, separately. The USV achieved a satisfactory path-following performance under the proposed controller. Figure 5 depicts the curves of the yaw angle error $\varphi_e$ and the velocity error $u_e$ under different disturbances. From Figure 5a,b, we can learn that tracking errors $\varphi_e$ and $u_e$ can all converge to a small neighborhood of zero in a short time and maintain a high robustness during system steady tracking stages. Figure 6 shows the time evolution of control inputs, and the norm of the adaptive update weight matrix is presented in Figure 7. By observing these results, we can learn that all of these closed-loop signals are bounded.

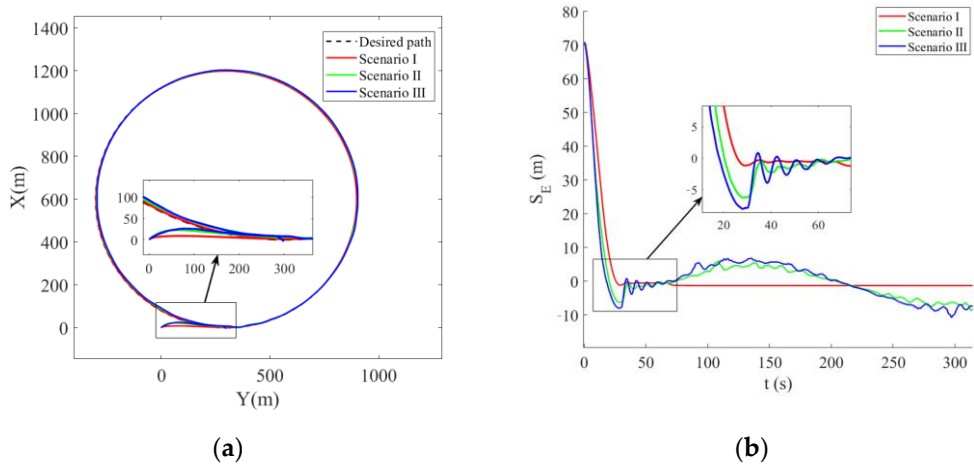

(a)  (b)

**Figure 4.** Time responses of position trajectory and tracking error: (**a**) position trajectory and (**b**) transverse deviation $S_e$.

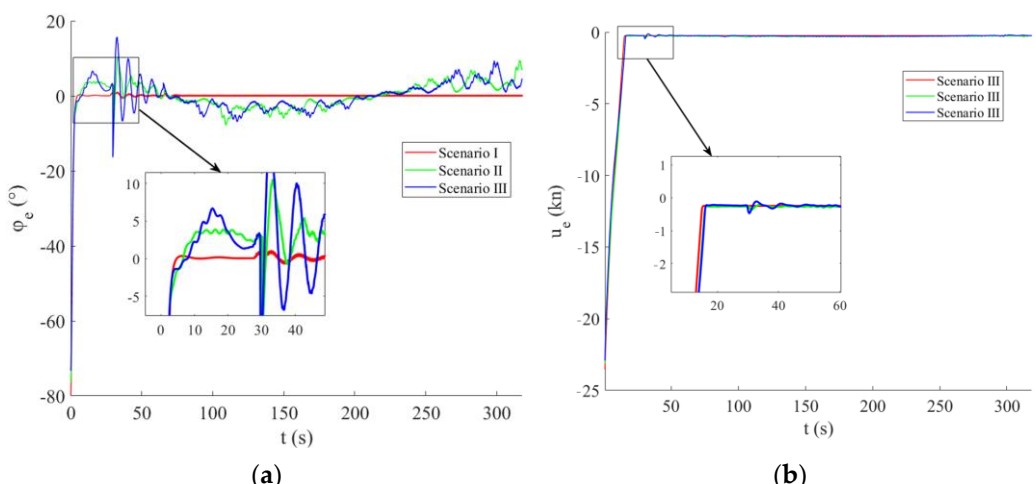

**Figure 5.** Time responses of tracking errors: (**a**) the yaw angle error $\varphi_e$ and (**b**) the velocity error $u_e$.

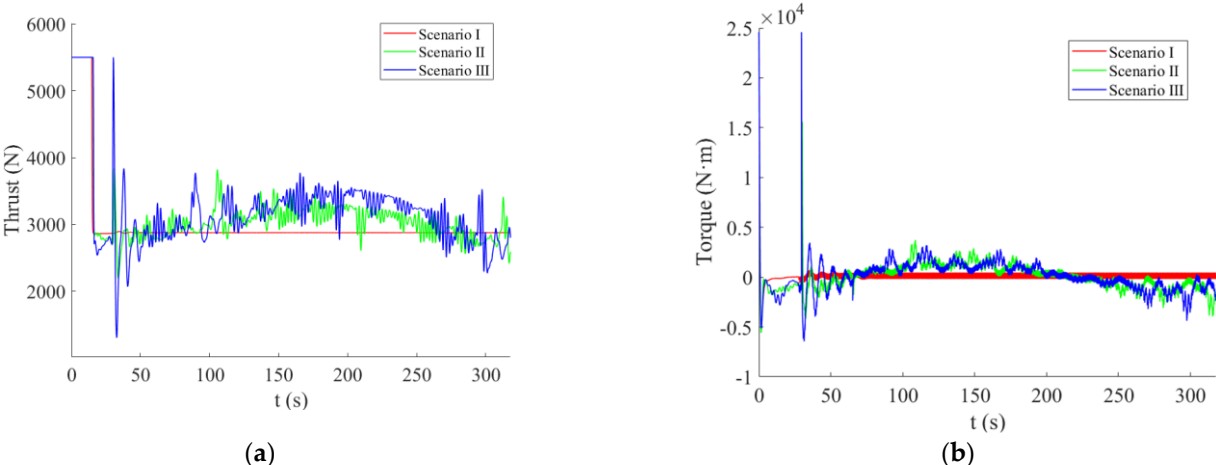

**Figure 6.** Time evolution of control inputs under different scenarios: (**a**) the control force $\tau_u$ in surge channel and (**b**) the control torque $\tau_r$ in the yaw channel.

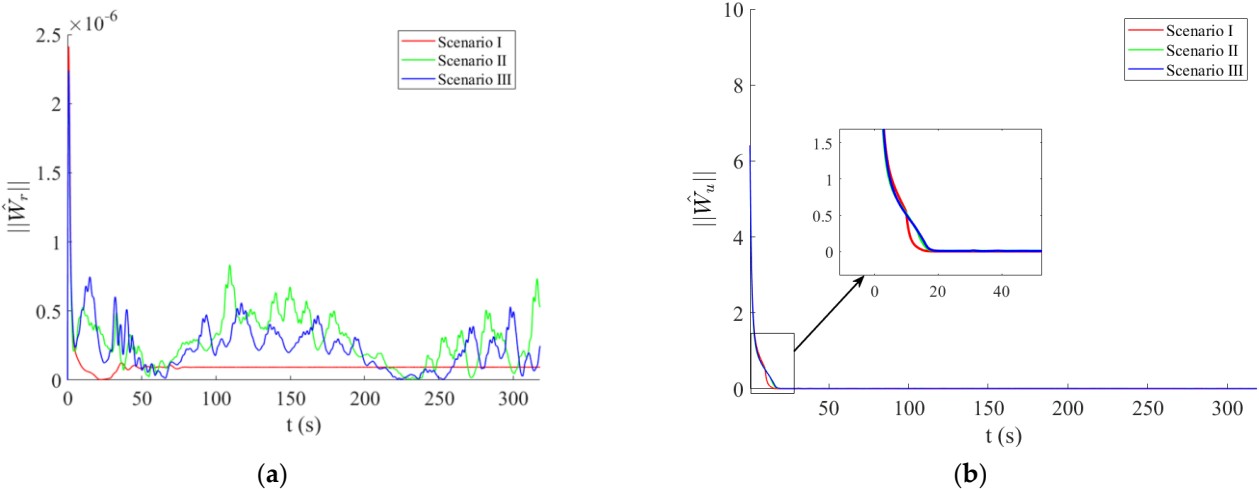

**Figure 7.** The norm of the adaptive update weight matrix under different scenarios: (**a**) the adaptive update $\|\hat{W}_r\|$ and (**b**) the adaptive update $\|\hat{W}_u\|$.

Under Scenario I, due to the absence of interference from the external disturbance, the fluctuation of the transverse deviation is small, and the yaw and velocity can quickly

respond to the expected value, which has a good tracking effect. Under Scenario II, environmental factors have a certain impact on the movement of the USV. There is a small overshoot and tracking deviation when the USV enters the circular path, but it soon tends to stabilize. The velocity error is stable around a value of zero, and the yaw error fluctuates 10 degrees up and down around the zero value. The circular path tracking effect is good. Under Scenario III, due to the harsher sea conditions, the influence of external environmental factors on the USV movement is more obvious. With the increase of velocity, the USV crosses the resistance peak area, and the violent model perturbation makes the USV have a certain overshoot and a tracking error of 20 m when it first enters the circular path. However, under the adjustment of the guidance law and controller, USV can still follow the desired path more accurately, and the tracking effect is tolerable. In summary, the controller shows desirable robustness under ocean disturbances.

## 6. Conclusions

This study describes a neural network-based adaptive sigmoid circular path-following control system for underactuated unmanned surface vessels under ocean disturbances. To facilitate the circular path guidance objective, the look-ahead angle is introduced into the guidance law to determine the LOS point of the circular path. Second, to enhance the guidance performance under external disturbances and parameter adaptability under different surge speeds, a sigmoid function-based compensator is subsequently constructed in the guidance law. Then, using the neural networks, an adaptive dynamic tracking controller is designed for the USV to realize the yaw angle tracking and speed tracking objective. Finally, a set of simulation verifications are performed under different ocean disturbances. The simulation results reveal that the presented scheme can effectively realize the path-following control objective. In addition, these results also illustrate the excellent control performance and robustness of the proposed control schemes with external disturbances.

**Author Contributions:** Conceptualization, Y.R., L.Z. and W.H.; software, Y.R. and L.Z.; validation, Y.R., L.Z. and X.C.; investigation, Y.R. and L.Z.; methodology, Y.R. and L.Z.; data curation, W.H. and X.C.; writing—original draft preparation, Y.R.; writing—review and editing, Y.R. and L.Z.; visualization, Y.R. and X.C. All authors have read and agreed to the published version of the manuscript.

**Funding:** This research was funded by Excellent Youth Foundation of Heilongjiang Province of China (grant number YQ2021E013) and The National Key Research and Development Program of China (grant number 2021YFC2803400).

**Institutional Review Board Statement:** Not applicable.

**Informed Consent Statement:** Not applicable.

**Data Availability Statement:** Access to the data will be considered by the authors upon request.

**Conflicts of Interest:** The authors declare no conflict of interest.

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
