# Peer review of "Neural Network-Based Adaptive Sigmoid Circular Path-Following Control for Underactuated Unmanned Surface Vessels under Ocean Disturbances"

_jmse, doi:10.3390/jmse11112160_

Round 1
Reviewer 1 Report (Previous Reviewer 2)
Comments and Suggestions for Authors
REPORT ON THE REVISED ARTICLE with title “Neural network-based adaptive proportion circular path-following control for under actuated unmanned surface vessels under ocean disturbances”
Manuscript ID: jmse-2563666
Authors: Yi Ren , Lei Zhang , Wenbin Huang , Xi Chen
Submitted Journal: Journal of Marine Science and Engineering
EARLIER SUGGESTIONS and PRESENT RESPONSES
SUGGESTION-1)The similarity of the article is 34%. It is a little high. See, the attached PDF file.
5% of the article was take from “Bin Zhou, Ziyang Huang, Bing Huang, Yumin Su, Cheng Zhu
Dynamic event-triggered trajectory tracking control for under actuated marine surface vessels with positive minimum inter-event time guarantees
Ocean EngineeringVolume 263, 1 November 2022, 112344”
and 5% was taken from another paper.
These rates should be reduced lower rates.
RESPONSE: The similarity rate is now 28%. It was reduced from 34% to 28%. Please, see, the attached PDF file as the similarity report. It has been reduced accordingly, and it is accetebale.
SUGGESTION-2)All parameters of the system (1) can be defined such that m_11,…,m_33 and some other can be different from the zero.
RESPONSE: They have been defined accordingly and marked on PDF file. See, the page 4.
SUGGESTION-3) All parameters of the Lyapunov function (34) can be defined clearly such that this function has to be positive definite.
RESPONSE: All parameters of the Lyapunov function (34) have been defined accordingly and they have been marked on PDF file. See, page 14.
SUGGESTION-4)The proof of uniformly ultimate boundedness is not clear, and it is also not satisfactory, too. It can be more clear and satisfactory.
RESPONSE: New information have been added to the proof. Now , the proof seems better than before. See, the pages 11 and 12, the marked spaces.
SUGGESTION-15) There are several minor typos, punctuation errors and grammatical mistakes. The article can be carefully checked and the necessary minor corrections can be done.
Some references of the article are not regular. They can be re-arranged as regular.
RESPONSE: Some revisions have been done accordingly. During the possible galley proofs, English language are checked.
SUGGESTION-6) The following related papers of stability and boundedness of solutions of higher order differential equations can be added to the references of this manuscript to update them:
RESPONSE:They have been added accordingly.
According to the above response, I would like to suggest the acceptation of the REVISED article in “Journal of Marine Science and Engineering” .
Comments on the Quality of English LanguagePlease, see, the report.
Author Response
Please see the attachment

Reviewer 2 Report (Previous Reviewer 3)
Comments and Suggestions for Authors
The manuscript has been significantly improved and now warrants publication in JMSE, further revisions are not required.
Author Response
Please see the attachment.

Reviewer 3 Report (New Reviewer)
Comments and Suggestions for Authors
The manuscript requires very extensive revision to eliminate the numerous typographical, grammatical and editorial errors indicated on the annotated manuscript. More detailed information should be provided how the guidance and the controller parameters are chosen. The parameters of the RBFNN used in the simulation must also be thoroughly discussed. In view of this, I would recommend acceptance of the manuscript subject to major revision.

The manuscript requires very extensive revision to eliminate the numerous typographical, grammatical and editorial errors indicated on the annotated manuscript.
Author Response
Please see the attachment

Reviewer 4 Report (New Reviewer)
Comments and Suggestions for Authors
This work demonstrates a commendable effort in addressing the challenging problem of path-following control for underactuated unmanned surface vessels (USVs). The authors have presented a well-structured and comprehensive study, proposing a novel neural network-based adaptive sigmoid circular path-following control scheme for underactuated USVs operating under ocean disturbances. The paper effectively highlights its contributions, provides thorough explanations of the methodology employed, and offers valuable simulation results to support the claims.
The clear organization of the paper, from problem statement to methodology and simulations, makes it accessible to readers. Furthermore, the practical implications of the proposed approach in marine missions, such as environmental monitoring, ocean exploration, and military tasks, add significant value to the research.
Overall, this is a successful work. Nevertheless, as a reviewer, I would like to offer some minor suggestions to the authors.
1. While the paper mentions addressing unmodeled dynamics in introduction, it would be helpful to briefly explain what these unmodeled dynamics might be. Providing examples or a bit more context could assist readers in understanding the challenges being tackled (optional).
2. While the introduction briefly discusses the significance of path-following control, it could benefit from a more detailed explanation of why precise path-following control is crucial in various applications. This could help underscore the practical relevance of the research.
3. Mentioning specific real-world applications where the proposed control system might be used would add practical relevance to the introduction. For example, elaborating on how the technology could benefit environmental monitoring, ocean exploration, or military tasks would help readers connect with the research (optional).
4. Please check all figures. Figures should be mentioned in the text. For example, Figure 2 is not mentioned anywhere in the text. Additionally, Figure 5, both parts (a) and (b), remain unmentioned. These aspects should be addressed to ensure better reader comprehension.
5. Emphasize simulation results in conclusion. For instance, you can say something like: “The results of our comprehensive simulations, conducted under various ocean disturbance scenarios, demonstrate the effectiveness of the proposed circular path-following control scheme. These simulations have shown that our approach maintains excellent path-following capabilities while robustly handling external disturbances.”
Author Response
Please see the attachment

This manuscript is a resubmission of an earlier submission. The following is a list of the peer review reports and author responses from that submission.
Round 1
Reviewer 1 Report
Comments and Suggestions for Authors
The paper presents a neural network-based motion control of an unmanned surface vessel based on a line of sight (LoS) approach. Although it is interesting, the proposed problem is quite old and there have been many attempts made in the past two decades. The reader could not find the original contributions of the paper/authors, the proposed combination is one of the well-known approaches. Further, the demonstration of the effectiveness of the scheme is limited to computer-based simulations that too very simple trajectories, the robustness of the proposed scheme is not investigated. How the proposed scheme is better than the existing schemes not provided anywhere.
There selection of several parameters is not described anywhere in the paper.
Comments on the Quality of English LanguageThere are several typos and grammatical errors, so it is better to perform thorough proofreading.
Reviewer 2 Report
Comments and Suggestions for Authors
Article Title: Neural network-based adaptive proportion circular path-following control for under actuated unmanned surface vessels under ocean disturbances
Manuscript ID: jmse-2563666
Authors: Yi Ren , Lei Zhang , Wenbin Huang , Xi Chen
Submitted Journal: Journal of Marine Science and Engineering
In this article, a circular curve path following controller for an under actuated unmanned surface vessel (USV) suffering from unmodeled dynamics and external disturbances is designed.
Here, from the MATHEMATICAL VIEW, the motions of the USV are defined in the horizontal plane. The model of the USV is described by the differential system (1). Next, the differential system (1) is simplified as the differential system (2).
- Theorem 1 has sufficient conditions such that the tracking errors are uniformly ultimate bounded and the closed-loop system is asymptotically stable. The proof this theorem is done by the Lyapunov’s second method via a Lyapunov function and La Salle’s invariance principle.
Here, simulations verification are provided
This is a very interesting article such that it includes mathematical model and applications.
The asymptotic stability result s is corrected.
However, to the best information of this referee, from the mathematical view, this article needs REVISION:
SUGGESTIONS for REVISIONS
1)The similarity of the article is 34%. It is a little high. See, the attached PDF file.
5% of the article was take from “Bin Zhou, Ziyang Huang, Bing Huang, Yumin Su, Cheng Zhu
Dynamic event-triggered trajectory tracking control for under actuated marine surface vessels with positive minimum inter-event time guarantees
Ocean EngineeringVolume 263, 1 November 2022, 112344”
and 5% was taken from another paper.
These rates should be reduced lower rates.
2)All parameters of the system (1) can be defined such that m_11,…,m_33 and some other can be different from the zero.
3) All parameters of the Lyapunov function (34) can be defined clearly such that this function has to be positive definite.
4)The proof of uniformly ultimate boundedness is not clear, and it is also not satisfactory, too.
It can be more clear and satisfactory.
5) There are several minor typos, punctuation errors and grammatical mistakes. The article can be carefully checked and the necessary minor corrections can be done.
Some references of the article are not regular. They can be re-arranged as regular.
6) The following related papers of stability and boundedness of solutions of higher order differential equations can be added to the references of this manuscript to update them:
-On existence and continuity results of solution for multi-time scale fractional stochastic differential equation. Qual. Theory Dyn. Syst. 22 (2023), no. 2, Paper No. 49, 23 pp.
- Some stability and boundedness conditions for non-autonomous differential equations with deviating arguments. Electron. J. Qual. Theory Differ. Equ. 2010, No. 1, 12 pp.
I would like to suggest the acceptation of this article in “Journal of Marine Science and Engineering” after a satisfactory revision can be done.

See, the report.
Reviewer 3 Report
Comments and Suggestions for Authors
My comments are given in the attached file!
